# Deep Brain Stimulation: In Search of Reliable Instruments for Assessing Complex Personality-Related Changes

**DOI:** 10.3390/brainsci6030040

**Published:** 2016-09-07

**Authors:** Christian Ineichen, Heide Baumann-Vogel, Markus Christen

**Affiliations:** 1Institute of Biomedical Ethics and History of Medicine, University of Zurich, Winterthurerstrasse 30, Zurich 8006, Switzerland; christen@ethik.uzh.ch; 2Department of Neurology, University Hospital Zurich, Frauenklinikstrasse 26, Zurich 8091, Switzerland; Heide.baumann-vogel@usz.ch

**Keywords:** deep brain stimulation, personality, instruments, competencies

## Abstract

During the last 25 years, more than 100,000 patients have been treated with Deep Brain Stimulation (DBS). While human clinical and animal preclinical research has shed light on the complex brain-signaling disturbances that underpin e.g., Parkinson’s disease (PD), less information is available when it comes to complex psychosocial changes following DBS interventions. In this contribution, we propose to more thoroughly investigate complex personality-related changes following deep brain stimulation through refined and reliable instruments in order to help patients and their relatives in the post-surgery phase. By pursuing this goal, we first outline the clinical importance DBS has attained followed by discussing problematic and undesired non-motor problems that accompany some DBS interventions. After providing a brief definition of complex changes, we move on by outlining the measurement problem complex changes relating to non-motor symptoms currently are associated with. The latter circumstance substantiates the need for refined instruments that are able to validly assess personality-related changes. After providing a brief paragraph with regard to conceptions of personality, we argue that the latter is significantly influenced by certain competencies which themselves currently play only a tangential role in the clinical DBS-discourse. Increasing awareness of the latter circumstance is crucial in the context of DBS because it could illuminate a link between competencies and the emergence of personality-related changes, such as new-onset impulse control disorders that have relevance for patients and their relatives. Finally, we elaborate on the field of application of instruments that are able to measure personality-related changes.

## 1. Introduction

Deep Brain Stimulation (DBS) is a neurosurgical intervention that involves electrode implantation to apply electrical currents to target structures aiming at alleviating symptoms. More precisely, the surgical method involves a stereotactical implantation of usually quadripolar electrodes including an extracerebral “pacemaker” that modulates the activity of selected regions in the brain with electric impulses. The key advantages of this procedure are (1) its potential reversibility and (2) the possibility to postoperatively optimize treatment effects via an external programming device. So far, a great number of patients suffering from various neurological and neuropsychiatric disorders have been treated with DBS. DBS unquestionably is a remarkable therapy that has provided hope for many patients and that has been shown to be more effective than best medical treatment for some disorders. Among those, patients suffering from Parkinson’s disease (PD) and who are refractory to drug treatment represent by far the largest patient group.

In the meantime, the rapid development reached a non-undisputed broadening of the therapeutic spectrum [1]. DBS of the subthalamic nucleus (STN) has been established in randomized, controlled trials as an effective therapy for the motor symptoms of PD [2,3,4] and, consequently, the number of patients being treated by DBS has steadily increased. Along this increase, challenges arose with regard to appropriate patient selection and side-effects, to name a few. With some delay, the systematic investigation of neuropsychiatric changes observed in patients treated with DBS for movement disorders found their way into the scientific literature, first as anecdotal reports and later in the form of quantitative research studies [5]. 

In sum, DBS has demonstrated dramatic symptom relief for a multitude of patients. However, complex non-motor changes following DBS interventions have been described. Because there is only a very limited number of instruments that are able to validly measure complex personality-related changes, there is great need for the development of new and reliable instruments in order to collect information and to evaluate these changes. As will be seen, there is a need to more thoroughly explore e.g., morally relevant behaviours (such as impulse control disorders, ICDs) with a particular emphasis on psychosocial competencies. The underlying competencies that might be dysfunctional secondary to disease, pharmacological therapy or neuromodulation interventions aiming to treat patients suffering from diseases, however, are hardly the focus of current DBS-research. In turn, new instruments that are able to quantify and depict such competencies might be highly relevant because they can yield explanatory power regarding psychosocial changes that are decisive for patients and their relatives.

## 2. Complex Changes after DBS Interventions

### 2.1. Non-Motor Problems Following DBS in Movement Disorders

In what will follow and for the sake of clarity, we restrict the argumentation of this contribution to the context of movement disorders and especially PD, even though we are aware of the fact that strictly speaking, PD has a well-documented neuropsychiatric impact on patients. Because STN-DBS in PD is, apart from stimulation of the globus pallidus internus (GPi), most frequently performed, the argumentation below takes up studies that investigated basal-ganglia dysfunction. Furthermore, a recent study provides Class II evidence that STN DBS offers more off-phase motor improvement than GPi DBS with similar risk for behavioural, affective and cognitive complications [6].

Whilst DBS aims primarily at improving motor symptoms in PD, accumulating knowledge points toward non-motor complications. Because a large number of fibers converge in the basal ganglia nuclei on a very small area, it is not surprising that targeting specific functions and manipulating them in an isolated fashion is tremendously difficult. In fact, intervening into basal ganglia physiology bears the risk of modulating non-motor functions [7,8,9]. The difficulty of specific targeting is even greater when factoring in recent notions of re-entrant or interconnected cortico-striato-pallido-thalamo-cortical loops representing different frequencies [10]. In fact, by manipulating a specific node in the network, one might influence as many different functions, depending on the degree of shared functionality with other circuits. Hence, DBS intervenes with a very complex network [11], the shear complexity of which has probably only started to be deciphered.

A large body of evidence implicates the role of the basal ganglia (BG) in the processing of non-motor signals and several psychiatric disorders such as schizophrenia, obsessive compulsive disorder (OCD), phobias and panic attacks, depressive states, addiction and eating disorders [12,13,14]. Similarly, neuropsychological changes in humans following DBS interventions have been observed [15], denoting the characterization of the BG as centers of convergence encoding motor, cognitive, associative and affective processes. Notably, the STN, one of the most commonly targeted structures for DBS in PD, has a strategic position due to its connections to both BG output structures (the GPi and substantia nigra pars reticulata (SNr)). Unsurprisingly, modulation of STN-signaling has therefore demonstrated to result in impulsive responding and dysfunctional inhibitory control, such as perseveration, obsessions and compulsions [16]. Hence, it is very well possible that interference with basal ganglia nuclei and the STN specifically, by disease or interventions, can modulate associative and limbic processing. Notably, interventions not only include DBS but also e.g., pharmacological treatment. The latter has demonstrated the potential of causing unintended side-effects either in combination with DBS or by itself e.g., when drugs are reduced too promptly after DBS initiation or in case of dopamine agonists [17]. In addition, the dopamine agonist withdrawal syndrome (DAWS) and dopamine dysregulation syndrome (DDS) leading to neuropsychiatric symptoms and decreased self-control, respectively, have been described in PD. Therefore, it is widely appreciated that pathological processes and pharmacological treatment alike can lead to alterations in the processing of emotional, cognitive and behavioural stimuli [18]. 

More generally, a number of behavioural and affective sequelae, such as hypomania, new onset impulse control disorders (ICDs including hypersexuality, pathological buying, pathological gambling, and addiction to levodopa), logorrhea, irritability, impatience and aggression, distractibility and attention problems, egocentrism, obstinacy, and lying have been described following DBS treatment in humans [15,16,19,20]. In the meanwhile, there is also evidence for changes that can be evaluated positively such as increased emotional wellbeing that results in increased Quality of Life (QoL) (see Section 2.2).

Undisputedly, complex non-motor changes have been described following DBS. In this contribution, we extend the topic of non-motor problems and deliberate on complex, non-motor changes for which the evaluation is unclear (i.e., such changes are not per se problematic and can even be positively evaluated). Patients who experience substantial symptom relief may develop new interests and behave differently. While the evaluation of whether these changes are problematic is important, we may first have to make sure that complex changes can reliably be measured. This includes the possibility that strictly differentiating between the measurement and evaluation process is eventually not possible, in particular when measuring personality changes where a positive or negative evaluation could be intrinsic to the measurement process. Nevertheless, and as will be seen, sensitive instruments are highly needed. First, however, we will investigate the nature of complex changes and the fundamental problem they are associated with.

### 2.2. Complex Changes: In Search of Reliable Instruments

Complex changes can be described as side-effects characterized by two gradual, qualitatively described dimensions. They include measurement complexity of side-effect on the one side and relative life impact of the side-effect weighted by its incidence in the natural disease history on the other side (see [20], Figure 2, for detailed information). Hence, complex changes represent side-effects that are associated with a high level of measurement complexity—an indirect evidence for it being an above average variance of the documented prevalence of a specific side-effect—and a correspondingly high level of relative life impact for the patient but also his/her social surrounding. Unfortunately, our own research provides evidence that the third-person perspective is almost never assessed in current practice (the usage of test scores that emerge from persons affiliated with the patient, are basically nonexistent [21], with the exception of some few recent studies (e.g., [22,23]). Paradigmatic examples of complex changes include changes in personality and moral behaviour. Because there is the problem of measurement complexity, predicting side-effects relating to psychosocial functioning of the patient is currently difficult. Despite the fact that patients raise their concern over the propensity of DBS to cause personality-related changes and that cases in which sudden alterations of personality after DBS have been described (see next paragraph), less emphasis has been put on the construction of instruments for quantifying personality-related changes. It thus stands to reason that complex changes in general (i.e., not limited to stimulation-induced changes) are generally underreported and hence affect the patient population to a much greater extent than assumed. The relative life-impact, in the meanwhile, depends on not only the type of side-effect but also on factors such as, among others, the pre-operative psychosocial status of the individual and premorbid personality traits.

As listed above, non-motor problems following DBS interventions—factors that clearly constrain the effectiveness of this type of intervention—often comprise a neuropsychiatric dimension. While some studies investigated non-motor problems following DBS interventions, few if any have focused on more demanding notions of personality and psychosocial competency [15], probably because of the complexity of the subject matter. For example, STN-DBS has been associated with deficits on a variety of tasks that require inhibition of prepotent responses and response selection during situations of high conflict (for a review, see [16]; for augmented impulsivity, see [24,25,26]; for a study demonstrating increased impulsivity assessed by the Barratt Impulsiveness Scale (BIS), see [27]), but few investigations included more profound notions of personality (see next paragraph). With regard to changes in mood and behaviour, a meta-analysis involving 1398 patients and 82 studies by Temel et al. [28] outlined that 8% suffer from depression, while activities of daily living (ADL) score improved by 52%, consistent with other reports documenting an improvement in QoL that is only related to physical aspects but not to mood ([29]: prospective study with non-implanted PD patients as control group). The latter study also highlighted 9% psychiatric complications (compared to 3% in the control group). Another more recent study found few changes in mood and behaviour with unilateral STN or GPi DBS, relating to worsened anxiety, depression and mania [30]. Other studies also revealed an increase in QoL including emotional well-being (for improvements of anxiety and depression see [31,32,33], in case of stigma and bodily discomfort see [34,35]). Particularly, the study of Witt and colleagues [19] found an improvement in anxiety, otherwise psychiatric adverse events in 16.7% of patients (compared to 12.7% for best medical treatment group) and a decrease in frontal cognitive functioning with no consequences on improvements in QoL. Finally, alterations in decision-making of PD patients measured by the Iowa gambling task (IGT) have been demonstrated by Pagonabarraga and colleagues [36] (with results pointing at similar decision-making deficits as seen in ventromedial prefrontal cortex (PFC) lesioned patients and pathological gamblers). Even though incorporating only a small sample size, Bentrup et al. [37] found an increase in “novelty seeking” in two out of 15 patients besides a decrease of sociomoral judgment on the six-level Kohlberg scale. On the contrary, Brandt et al. found that DBS may temper the tendency of risk-taking on risky decision making tasks with DBS patients being more risk-averse in ambiguous-risk situations [38].

With regard to personality changes mirroring temperament and character, Houeto and colleagues [39] have examined, in an earlier study and retrospectively, adjustment disorders (using the social adjustment scale (SAS)), personality changes (using the Iowa rating scale of personality change (IRSPC)) and psychiatric disorders (using psychiatric interviews and the mini international neuropsychiatric inventory). Results indicated moderately to severely impaired social adjustment by 62.5% while personality traits were improved by 35% and aggravated by another 35%. Table 1 lists these and the forthcoming outcomes of some recently performed DBS-studies together with a brief test description, overall revealing that more studies and new instruments would be desirable. Another retrospective study of the same researchers [34], however, observed very different results including unmodified personality traits (as assessed by the Temperament and Character Inventory-revised (TCI-R) and this contrary to [40], who found increased scores on two Novelty-Seeking subscales of the same measure) and social adjustment apart from improved depression, anxiety and QoL. Denheyer and colleagues [41], on the other hand, used the Frontal Systems Behavior Scale (FrSBE) in order to assess behavioural changes including apathy, disinhibition and executive dysfunction. In a retrospective study with a non-representative sample, all scores increased significantly. Notably, most of the above-listed instruments have important limitations, such as, for example, the subjective nature of the FrSBE that is influenced by e.g., preconceived expectations about the outcome of DBS. Taken together, these outcomes imply relatively contradictory results. To be clear, psychosocial dysfunction and changes relating to altered character after DBS, potentially resulting in difficulties of social adjustment, satisfaction gaps and conflicting outcome interpretations between patients, their relatives and practitioners, have vaguely been described ([42,43], or studies using e.g., the SAS), but have not been investigated focusing on causal elements leading to such changes. Because few empirical studies that investigate personality changes exist, recently Lewis and colleagues [22] have examined the latter by use of semi-structured interviews and a neuropsychiatric battery (Parkinson’s disease questionnaire-PDQ-39, Beck depression inventory (BDI-II), apathy evaluation scale (AES), state-trait anxiety inventory (STAI-state), self-report manic inventory (SRMI), Barratt impulsiveness scale (BIS-11), hypomanic personality scale (HPS) and mini mental status examination (MMSE)), highlighting that personality changes occur between 22% (self-evaluation) and 44% (evaluation by caregiver, e.g., spouses), with another 57% perceiving mood changes as positively, thereby emphasizing the relevance of such investigations. However, higher apathy and anxiety levels were found in the negative change group. The fact that the used standard measurement scales were unsuccessful in adequately reflecting personality and mood changes in this study, substantiates the need for better and refined instruments. One more recent study investigated personality changes after DBS [23]: the 125-item version of the TCI (TCI-125), the urgency-premeditation-perseverance-sensation seeking (UPPS) impulsive behaviour scale and the Eysenck Personality Questionnaire (EPQ) have been used with findings relating to increased impulsivity and personality changes in Persistence- and Self-Transcendence test scores (see Table 1). Notably, the previously listed non-motor problems are likely to be associated with other fine-grained changes which may reach far into the domain of personal convictions, values and sensitivities. These changes might be so nuanced that they will slip through current assessment of psychiatric test batteries (for example, see [22]). Hence, even though there are a number of tests for investigating e.g., impulsivity (e.g., Eriksen flanker and Simon task, the Stroop color word interference task and random number generation; [16]), less emphasis has been put on the construction of instruments for quantifying nuanced personality-related changes [5,44], including instruments depicting personal competencies in sociomoral information processing in order to have instruments at one’s disposal that pick up relevant topics that matter to patients and caregivers.

Addressing complex personality-related changes with the requisite rigour may explain causal elements for the emergence of conflicting outcome interpretations and social maladjustment that are relevant for patients and their relatives. Clearly, such instruments need to rely on newer insights of psychological research (see Section 4.2). While in the large majority of patients, symptoms relating to dominant expressions of behavioural phenotypes such as impulse control disorders or hypomania can be controlled (either they vanish spontaneously or by adjusting stimulation and/or drug treatment), more subtle changes have rarely been addressed so far due to the lack of sensitive instruments that measure complex changes beyond standard test-psychology. Given that some behavioural changes listed above had long-lasting social effects and damaged relationships that often only came to the fore through in-depth qualitative research [45], and given that patients express their concern over personality-related changes secondary to deep brain stimulation—as clinical experience at our clinic shows—together with the magnitude of the life impact for patients and their relatives, it is staggering that, if any, only a very limited set of data e.g., [22,23,46] and few established instruments apart from very general personality assessment tools of standard test psychology (e.g., the big five personality test, but see Table 1) are currently available that deal with the topic of personality-related changes. The limited data is surprising also when considering that changes in personality and mood under DBS in PD are discussed both in the clinical but also the ethical literature [47,48,49,50]. In addition to the problem of measuring complex changes, there is also the evaluation difficulty relating to the problem of how to evaluate such changes. As Kraemer [51] pointed out, “alienation from alienating conditions” can occur. The latter denotes the difficulty of how to evaluate changes in personality. Are marital problems following DBS implantations categorically social maladjustments or may they, in a proportion of patients, reflect a changed personality denoting to more fundamental desires and thoughts that are at the core of the patients’ true self? Does a given change in personality symbolize alienation or approach to a patient’s pre-morbid personality? Without going into the details of such difficult questions, it is enough to stress that instruments and their ability to measure complex changes are a necessary precondition in order to move on and evaluate whether complex changes are problematic (including e.g., notions of felt-authenticity and felt-alienation). This includes the very possibility that not all changes are per se negative and that also positive personality changes following DBS surgery can occur. The latter is particularly important in the context of psychiatric DBS intervention where the positive change of ones’ personality is at the core of the therapeutic aim. These strategies may inform ethically responsible decision making in e.g., the referral practice of DBS interventions (see Section 4.3).

In sum, we argue that there is currently a lack of valid instruments that adequately depict changes in psychosocial processing. Even though a limited number of standardized questionnaires and tests are available, they may not reflect sufficiently the behavioural and affective changes and their effects in real life. Therefore, new avenues for the better description of complex personality-related changes that may explain causal elements for the emergence of conflicting outcome interpretations, in addition to social maladjustment that are relevant for patients and their relatives, need to involve instruments that rely on newer insights of psychological research.

## 3. Psychosocial Competencies

### 3.1. Individual Identity, Personality and Psychosocial Competencies: Conceptualizations and Interconnections

As proposed elsewhere [5], a conceptual clarification of individual identity and personality is decisive for evaluating and measuring potential personality-related changes in the future. Even though the focus of this contribution is empirical rather than conceptual, a brief conceptual clarification is necessary. While “individual identity” may be understood as a philosophical concept, “personality” refers to a psychological one [5]. Because the latter is empirical in nature, it is part of this work. Personality can briefly be described as the combination of certain characteristics or qualities that form an individual’s idiosyncratic character. It is commonly defined as “the organized set of characteristics possessed by a person that uniquely influences his or her cognitions, motivations, and behaviours in various situations” [52]. Psychosocial competencies are one class of examples that can influence and guide a person’s cognition, motivation and behaviour and hence align with the previous definition of personality. They include, among others, self-regulatory skills, the ability to identify issues linked to personal desires and values and to align one’s behaviour according to one’s self-conception, the desire to orient oneself towards and strive for one’s ideals, skills to resolve conflicting (internal or external) tendencies and the ability to act consistently with one’s internal thoughts and ideas. They furthermore guide human cognition through schemas and scripts (i.e., cognitive representations, e.g., [53,54]). Personality changes as understood in current psychology refer to alterations in the “Big Five” personality traits (i.e., extraversion, neuroticism, agreeableness, conscientiousness, openness to experience; see [55]), representing a very vague sum of a set of traits that can be altered. Notably, the latter concept has recently been expanded by the HEXACO model that adds a sixth trait circumscribing ones’ personality. That this sixth personality trait is precisely moral in nature (honesty-humility-dimension) is certainly an interesting development that aligns with the requested emphasis on morality and its associated competencies expressed in this contribution (see Section 4.1).

While a whole plethora of different competencies are necessary and amalgamate in interpersonal human conduct, the time is rife for investigating basic mechanisms of psychosocial competencies and to start developing instruments for measuring the underlying competencies. The fact that psychological competencies are needed in situations of complex decision making and behaviour, together with the likely potential of DBS to influence such competencies, corroborates the need of instruments to document and evaluate changes of psychosocial functioning in order to better support patients and their social surrounding. Moreover, DBS provides a possibility to investigate changes depending on type of stimulation and anatomical target.

### 3.2. Human Behaviour as an Expression of Social Competencies

With the emergence of psychology as a scientific discipline, questions arose regarding the identification of psychological skills that are necessary for social interaction but also regarding the extent and interplay of biological determinants that affect psychological processes. Broadly speaking, human behaviour includes processing external and internal stimuli and interacting with the external world. Hence, an interactionist view is mandatory in order to explain and understand behaviour. 

The same holds with regard to moral behaviour where analogously personal and environmental or contextual factors operate interactively in determining behaviour. Personal factors include a specific set of competencies while environmental factors involve conditions promoting or obscuring moral conduct (e.g., through priming by family pictures, time pressure, economic incentives, [56,57,58,59]). In addition, moral behaviour involves a normative reference frame to which the subject has at least partial access [60]. Thus, apart from environmental factors as well as ones’ personal moral identity, personal competencies significantly influence human action. Because it is likely that disorders and treatment approaches when interfering with the central nervous systems integrity directly or indirectly alike, might influence psychosocial competencies, the investigation of abilities of patients is genuinely pertinent. In line with this, it might be possible that DBS detrimentally influences a specific subset of competencies in a way that facilitates the emergence of behavioural disorders, such as ICDs, consequently leading to complex changes including e.g., psychosocial maladjustment. Unfortunately, the potentially dysfunctional competencies secondary to DBS interventions are hardly the focus of current research. So far, socio-moral behaviour and moral information processing following deep brain stimulation interventions has received little attention, even though problems in social adjustment raise questions that refer to psychosocial competences and abilities of the patient. New instruments that are able to quantify and depict such competencies might be highly relevant because they can yield explanatory power regarding psychosocial changes that are decisive for patients and their relatives.

## 4. Moral Psychological Competencies, Requirements for Instrument Implementation Strategies and Future Use of Instruments in DBS Research and Therapy

### 4.1. Moral Psychological Relevance for Assessing Complex Changes in Moral Information Processing as an Exemplary Case

The field with the richest empirical knowledge on how agents reason, decide and act morally is still moral psychology. Undisputedly, moral behaviour in everyday situations reposes critically on specific skills moral agents imperatively have to be equipped with. Many of those competencies and abilities are psychological in nature. Apart from earlier approaches (see e.g., [61]) Tanner and Christen [62] proposed an adapted framework termed “Moral Intelligence” that describes the process logic of moral behaviour by taking into account fundamental knowledge about implicit and explicit psychological processes and theoretical insights into morality, and superimposed on this, aiming at translating these processes into competencies. The framework includes a content-related component along a set of motivational, perceptive, decisional and action-related abilities. Besides that, progress in cognitive and social neuroscience has led to investigations of other important social constructs such as empathy and morality (see e.g., Jefferson Scale of Empathy, the multifaceted empathy test with one study revealing reduction of the “negativity bias” in patients with treatment-resistant depression by DBS [63], or the moral foundation scale by Haidt and the moral attentiveness scale by Reynolds). These and other strains of research, therefore, bear witness to the importance of identifying key competencies of human moral ability and how those competencies are rooted within and affected by psychological processes. Because moral competencies, such as the ability to recognize ethical issues in everyday situations, influences a person’s cognitions, motivations, and behaviours, changes in these competencies can lead to complex, personality-related changes. Other moral-related competencies include the possession of a more distinct desire to strive for moral goals, self-regulatory skills, the ability to resolve conflicting tendencies or being more prone to act consistently and courageously despite internal or external barriers. This rich variety of different skills that can be expressed in different degrees in part account for why people act uniquely and why they display unique personalities.

### 4.2. Thinking Ahead: Operationalization of Moral Competencies as an Exemplary Case

As an example, one particular psychological competency that is relevant in the context of moral behaviour is moral sensitivity (MS) [64], the ability to recognize (moral) issues in a given situation. Being conceptualizes as the first competency in the process logic of human moral behaviour (see Rest’s (1986) multi stage model of moral functioning [61] or the framework of Moral Intelligence [62]), moral sensitivity is an indispensable competence to enter decision-making processes and moral behaviour in general. Hypothetically, it might be possible that DBS constrains the sensitivity of an individual in such a way that makes it difficult for the patient to recognize that a given person might be harmed by certain actions, even though he is generally of the opinion that one should abstain from harming others. To be sure, changes in moral information processing are likely to occur also secondary to pathological processes and (e.g., pharmacological) treatment approaches. The use of refined instruments that depict neuropsychological competencies is, therefore, not limited to DBS. The decreased sensitivity to recognize e.g., the harming nature of certain actions following DBS intervention could symbolize a basis for explaining complex personality-related changes. Notably, there is a difference between the inability to subconsciously recognize that a given value might be harmed and the deliberate convictions somebody holds. While such fine-grained changes can sometimes be assessed by conducting qualitative semi-structured interviews in patients [20], the latter are often impracticable due to the time-consuming steps of post-coding. Often enough, in times of evidence-based medicine, and hence a quantitatively oriented medical discipline, effects are only becoming a relevant aspect of research if they can reliably be measured. Hence, quantitative instruments should supplement qualitative research, thereby emphasizing the prima facie importance of more vigorously investigating personality. Most importantly, and in order to safeguard clinical meaningfulness, abstract clinical scale-improvements have to be associated with an actual improvement in the individual patient’s life. While there were e.g., reports documenting measurable cognitive declines, the latter were found not to be very relevant for patients’ QoL. That the same holds for slight changes in personality is less likely, since changes in one’s personality are more probable to have an impact on daily life by being able to endanger relationships and family life [22]. They often affect interpersonal relations and inflict greater burden on caregivers [65], apart from the fact that changes can be subtle so that patients themselves may be unaware of them [66]. Statistically significant efficacy is, therefore, only a necessary, but not a sufficient condition, as it does not always correspond to meaningful changes. QoL assessments are one way to document clinical meaningfulness, but the construct itself is very difficult to assess (e.g., unrealistic expectations about DBS outcomes can fundamentally influence QoL in that no improvement is seen in QoL despite motor improvement). By aligning our own research to the mentioned need, we recently have built a computerized instrument to measure MS by taking into account recent insights from moral psychology [67]. Needless to say, instruments need to satisfy common psychodiagnostic standards including reliability, validity and other quality characteristics of psychological test-theory in order to guarantee that these instruments measure what they intend to measure and with the requisite precision.

Challenges of the instrument development process are multifaceted and include e.g., the incorporation of vague quantifiers in the context of psychosocial functioning. The step of specifying basic components is therefore utterly important. In addition, the delicate nature of the instruments’ content, covering morality including e.g., anti-social compulsions or sexual urges, poses implicit challenges to instrument development and data acquisition. Besides that, there might be the problem of overlapping functions denoting the difficulty of dissociating competencies that result in moral action. Instruments should likewise, to some degree, adhere to the criterion of generalizability or, as a minimum criterion, context-dependent instruments would need some form of justification. They are also expected to refrain from provoking biased responses by e.g., avoiding including the terms “ethical” and “moral”. In addition, instruments should take up recent insights of moral-psychological research. Correspondingly, instruments that are entirely based on self-reports are susceptible to bias, due to the provocation of social-desirable answer tendencies or e.g., reduced awareness based on frontal-subcortical circuit dysfunction. Finally, the methodological requirements should be as low as possible: because the categorization (post-coding) of issues (mentioned by participants) is time-consuming and requires the analysis of inter-rater-reliabilities, and because some implicit measurements impose participants to work on computers and in controlled environments, they are unsuited for quantitative research approaches.

In sum, in the context of neuropsychological competencies of socio-moral information processing, even fewer instruments exist at present. Future instruments that focus on socio-moral functioning should comply with the requirements of psychological test theory besides taking up recent insights of (moral) psychological research. 

### 4.3. Future Use and Advantage of Instruments Measuring Moral Competencies

Generally, the deployment of instruments that measure psychosocial functioning are conceivable for the referral practice in DBS interventions and as diagnostic outcome-measures in order to measure pre-post DBS effects. The prospective measurement of complex changes may help with making predictions regarding who is likely to experience clinically significant personality-related alterations, making individualized counselling possible, and aiming at minimizing negative impacts on patients and their families.

Because competencies are believed to be flexible entities that evolve and change over time, the focus on psychosocial competencies might encourage investigating means for modulating such competencies for the better, as a potential form of therapeutic interventions. By such an approach, subjects have the opportunity to inimitably learn more about as well as specifically train ones’ own competencies.

## 5. Conclusions

Whilst DBS has provided hope for a large number of patients and while a number of scales for e.g., assessing motor changes have been developed during the last years, there is a conspicuous lack of instruments that target and adequately depict personality-related changes specifically. Standardized questionnaires and tests are available, but they may not reflect sufficiently the behavioural changes and their effects in real life. Together with the highlighted lack of sensitive instruments, an appreciation of the concrete incidence of personality-related changes (evoked by DBS or other treatment approaches) is impossible. Therefore, there is great need for refined instruments that quantify complex, personality-related changes at a satisfactory level. While the clinical significance of any measured change has to be demonstrated, the frequency of such changes has to be investigated by representative samples and prospective study designs in order to systematically investigate these changes in patients relative to controls. Because such instruments may explain causal elements for the emergence of conflicting outcome interpretations and social maladjustment that are relevant for patients and their relatives, such research is desperately needed. Integrating caregivers and families’ perceptions of the patient and the impact on their life would complement this complex investigation, thereby safeguarding clinical meaningfulness. Given the challenge patients may face when finding a “new” personality after being freed from the motor symptoms, and the subsequent loss of (motor) autonomy that kept them fettered to PD, it is utterly important to give patients, their relatives and clinicians means for granting ways for measuring and assessing changes in an appropriate way. Hence, the time is rife for advancing DBS treatment along both the technological axis, and an axis that involves the holistic assessment including personality in its full intricacy in order to provide further help to many patients. Besides empirical research that includes the development of instruments and the systematic and longitudinal investigation of complex changes that relate to personality, conceptual investigation on identity and qualitative research with patients, their relatives and clinicians are highly relevant. It is only with the construction of sensitive instruments, allowing a subtle measurement of the type of change, that an evaluation of the positive or negative nature of change is possible. Finally, besides a clinical, it is also an ethical requirement to further investigate complex changes in order to responsibly apply DBS.

## Figures and Tables

**Table 1 brainsci-06-00040-t001:** Outline of some of the recently used measures of personality in deep brain stimulation (DBS) studies, including measurement description and study outcome with reference numbers in brackets.

Name of Test/Scale	Short-Description	Study & Main Study Results
Social adjustment scale (SAS)	Semistructured interview, performed in the presence of the spouse, that evaluates current social adjustment in terms of 44 items	Houeto et al., 2002 [39]: moderately to severely impaired social adjustment by 62.5%; Houeto et al., 2006 [34]: SAS global score and subscores (work, social life and leisure activities, family life, marital relations, and interaction with children) unmodified
Iowa rating scale of personality change (IRSPC)	30 characteristics are assessed, ratings are gathered from family members with regular contact with the patient	Houeto et al., 2002 [39]: personality traits were improved by 35% aggravated by 35% und unchanged by another 30%
Temperament and Character Inventory-revised (TCI-R)	Self-evaluation, four temperaments (Novelty Seeking (NS), Harm Avoidance (HA), Reward Dependence (RD), Persistence (PS)) and three characters (Self-Directedness (SD), Cooperativeness (CO), Self-Transcendence (ST))	Fassino et al., 2010 [40]: higher scores emerged on two Novelty-Seeking subscales; Houeto et al., 2006 [34]: unmodified personality traits; Pham et al., 2015 (TCI-125) [23]: patients reported lower score on the TCI Persistence and Self-Transcendence scales, after three months of subthalamic nucleus stimulation (STN-DBS), compared to baseline
Eysenck Personality Questionnaire (EPQ)	Self-report questionnaire: Extraversion, Neuroticism, Psychoticism and Lie scale	Pham et al., 2015 [23] (Neuroticism and Lie subscales (EPQ-N, EPQ-L)): no changes
Frontal Systems Behavior Scale (FrSBE)	Behavioral assessment of frontal lobe syndromes, includes items related to apathy, disinhibition, and executive dysfunction; 46-item behavior rating scale, self-evaluation and family evaluation	Denheyer et al., 2009 [41]: apathy, disinhibition and executive dysfunction increased
X	Semi-structured interviews, developed by the ELSA-DBS study group (a project that examines Ethical, Legal and Social Aspects of Deep Brain Stimulation with respect to health, quality of life and personal identity) to investigate motor, emotional, social, behavioural and cognitive functioning, activities of daily living and QoL	Lewis et al., 2015 [22]: personality changes occurred between 22% (self-evaluation) and 44% (evaluation by caregiver, e.g., spouses)

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
