# Peer review of "Deep Brain Stimulation: In Search of Reliable Instruments for Assessing Complex Personality-Related Changes"

_brainsci, 2016, doi:10.3390/brainsci6030040_

Round 1

Reviewer 1 Report

Thank you for the opportunity to review this very interesting paper. The authors argue for the need for better measures of personality and personality change in the context of DBS for the treatment of motor symptoms of PD. They discuss the importance of assessing moral sensitivity as one example. Such personality assessments need to reliable, valid, and patient friendly (i.e., easily administered).

The authors highlight a clear need in the literature. Greater assessment of personality and change is necessary. It will be particularly important to include relevant control groups (e.g., medication management versus DBS) to truly ascertain if any identified changes reflect DBS.

I have a couple of general broad comments regarding this manuscript.

First, I think that the authors imply that negative personality changes following DBS surgery are more common that has been my experience in caring for >2000 of these patients for 20 years. My experience has shown that it is far more likely to see significant personality changes associated with dopaminergic medications or ongoing disease progression than neurostimulation per se. This is not to dispute that some neurobehavioral changes may be evident following DBS in some, vulnerable patients, but we still don't have an accurate estimate of the proportion of patients who may experience such changes or the clinical meaningfulness of such changes. I recommend that the authors present a more balanced argument regarding the possibility of both negative and positive personality changes following DBS surgery. Once again, my concern is based primarily on clinical experience and discussions with several other clinicians who work with these patients daily. I recognize that this comment highlights the need for more detailed quantitative studies (supporting the authors' primary point) that, once again, should include careful discussion of the clinical meaningfulness of any reported statistical changes.

Second, there are at least two published papers which examined personality changes using reliable and valid measures following DBS surgery for treatment of PD. Both papers are limited by small sample sizes. (See Denhever, Kiss, Haffenden 2009. Neuropsychologia, 47(14)3203; Houeto et al (2002) Journal of Neurology, Neurosurgery, and Psychiatry 72: 701). There are existing measures that examine personality changes that may be highly relevant to patients and families. These include the Iowa Scales of Personality Change (Barrash) and the Frontal System Behavior Scales (Grace and Malloy).   Once again, the clinical significance of any measured changes needs to be clarified as well as the frequency of such changes in a large series of patients from a variety of DBS centers.

Finally, I am particularly intrigued by the authors' use of Moral Sensitivity as an example. They might want to cite the ever-growing literature on cognitive and social neuroscience which  examines the underlying neurobiology of important social constructs such as empathy and morality. This literature may provide other well validated measures that measure these important constructs. (As an aside the Jefferson Scale of Empathy has been administered with healthcare professionals. I don't know if it has been applied to patients with neurodegenerative disorders).

In summary, I believe that this is a very important topic and agree with the authors' primary thesis that we require reliable and valid tests of important and clinically meaningful personality characteristics. I encourage them to consider the possibility that the incidence of negative and enduring personality changes following DBS surgery may not be common and that positive changes may occur as well. Furthermore, statistically significant differences do not always correspond to clinically meaningful changes. This is a very important point. I prefer to see a more balanced paper. I encourage the authors to cite some of the papers that did examine personality change following DBS and note the existing measures that examine personality change. They may also want to consider the literature on cognitive and social neuroscience (cf Cacciopo).

Author Response

Please find attached the answers to the reviewers concerns.

Reviewer 2 Report

The authors focus on an area that is being to receive more attention - the effects of DBS on the psychological and personality related aspects of a person's functioning.  there are many examples in the literature of patients who have experienced changes following DBS leading to divorce, to admission to the hospital, to changes in behavior, etc.  However, there is no easy way to measure this, nor do we know whether there are ways to predict or anticipate who might experience these changes.  The authors make a call for new measures to capture these changes.  I agree with this argument.  I might suggest that the authors offer some suggestions for what might be currently available as proxies until new measures can be developed, as this could take some time.  are there any scales/tools that could be used in the interim?

Author Response

Please find attached the replies to the reviewers concerns.

Round 2

Reviewer 1 Report

Thank you for the opportunity to review this revised manuscript. It is improved and well worth publication.